# Effects of a One-Day Experiential Sheep-Rearing Experience on Motivation, Anxiety, and Frontal Lobe Brain Activity in Patients with Chronic Psychiatric Disorders: A Crossover Pilot Study

Nobuko Shimizu [1,*], Shingo Ohe [2], Keigo Asano [3] and Motohiko Ishida [4]

1 Faculty of Nursing, Toyama Prefectural University, Toyama-City 930-0975, Japan
2 Faculty of Nursing, Ishikawa Prefectural Nursing University, Kahoku-City 929-1210, Japan; ohe1111@ishikawa-nu.ac.jp
3 Faculty of Bioresources and Environmental Sciences, Ishikawa Prefectural University, Nonoichi-City 921-8836, Japan; asanok@ishikawa-pu.ac.jp
4 Scientific Cooperation Center for Industry Academia and Government, Ishikawa Prefectural University, Nonoichi-City 921-8836, Japan; moto@ishikawa-pu.ac.jp
* Correspondence: n-shimizu@pu-toyama.ac.jp

**Abstract:** The purpose of this study was to investigate the effects of a one-day sheep-rearing experience on motivation and anxiety levels in patients with chronic mental illness. The study assessed changes in oxytocin and cortisol levels and brain activity in the prefrontal cortex, which is known to be associated with emotion and motivation. The study employed a non-randomized controlled trial design, with participants receiving both an intervention day (sheep rearing) and a control day (usual psychiatric day care) in a crossover fashion. Written informed consent was obtained from all participants. The intervention day consisted of hands-on sheep rearing activities, while the control day consisted of general activities available at the psychiatric day care center. Results showed that the sheep-raising experience had an equal effect on motivation and increased mean oxytocin levels. In addition, significantly more activity was observed in the dorsolateral prefrontal cortex (DLPFC) region of the brain compared to typical psychiatric daycare activities ($p < 0.032$, $p < 0.043$). Participants tended to have increased oxytocin levels after sheep rearing, and the activation of the DLPFC has not previously been observed in animal intervention studies. These are new findings in psychiatric occupational therapy that may have effects on social cognition and interpersonal relationships in patients with chronic mental illness.

**Keywords:** sheep rearing; chronic psychiatric disorders; oxytocin; cortisol; anxiety; psychiatric daycare; apathy scale; near-infrared spectroscopy; brain activity; non-randomized controlled trial

## 1. Introduction

A survey conducted by the Cabinet Office (one of Japan's administrative agencies that is responsible for planning and coordinating important policies of the Cabinet) in 2018 revealed that many people with psychiatric disabilities are "isolated" from society, "withdrawn" at home, and have little contact with their families and communities [1]. Deinstitutionalization in Japan has been slow, and the number of psychiatric beds per 100,000 people remains the highest among Organization for Economic Cooperation and Development (OECD) countries at 269 beds [2]. However, intervention programs to improve mental health and prevent psychiatric disorders vary by country and age. In particular, patients with chronic schizophrenia, who are often hospitalized long-term in Japan, show negative symptoms, such as low motivation and fatigue, which hinder their return to society [3].

Schizophrenia develops during adolescence and causes dramatic and lifelong impairments in social and occupational functioning [4]. It is characterized by negative symptoms,

such as decreased motivation, spontaneous speech, and social withdrawal; positive symptoms, such as delusions and hallucinations; and cognitive symptoms, such as im-paired speech, attention, and thinking, resulting in impaired communication skills [4]. Barch proposed that dopamine dysfunction in schizophrenia results in decreased motivation and mobility [5]. In schizophrenia, the involvement of the prefrontal reward system (corticostriatal function) in cognitive control is considered deficient, resulting in reduced goal-oriented behavior [5]. Support that leads to an increased motivation and willingness of persons with psychiatric disorders and such disease characteristics is essential for promoting their reintegration into society. Many researchers are now attempting to improve social cognition in schizophrenic patients by conducting intervention studies using, for example, Wolwer et al.'s facial affect recognition (FAR) training program [6], the social cognition and interaction training developed by Penn et al. [7], and the social cognitive skills training developed by Horan et al. [8]. Some programs are more complex and eclectic, incorporating domains, or "broad-based" interventions, which encompass affective cognition, attributional styles, and Theory of Mind (ToM) treatment. Some interventions are brief, lasting one or two sessions (e.g., Combs et al. [9]), while others are sustained, lasting several months (e.g., Roberts and Penn [10]). They reported that the effects of social cognitive training techniques were related to social cognitive measures, but the effects on social cue perception and attributional style were not significant. In addition, these intervention techniques are interventions by psychologists, and it is difficult to provide interventions for a large number of people with disabilities one at a time in settings where there are few staff with specialized knowledge, such as in employment support settings. Therefore, it is important to develop alternative and adjunctive therapies to improve treatment outcomes in schizophrenia.

Animal therapies have been classified into animal-assisted activity (AAA) and animal-assisted therapy (AAT) and have been studied. AAA has been reported to have a positive effect on children's mental development, and AAT is effective for individuals with emotional disturbance and mental retardation. Recent studies have shown that AAT pro-grams are beneficial for patients undergoing treatment for schizophrenia [11–17]. However, these programs were implemented under certain conditions as part of treatment and rehabilitation, and most evaluation measures focused on symptom stability [17]. In addition, previous studies reporting the effects of AAT in patients with schizophrenia did not measure effects on oxytocin or cerebral blood flow.

The effects of animal interventions on humans can be roughly divided into physical, mental, and social; however, these effects do not occur in isolation. The brain mainly controls the mind and body. Therefore, the direct measurement of brain activity, the command center of behavior and mental activity, should search for the root of the effects on the mind and body. As of recently, near-infrared spectroscopy (NIRS) is now being used in the field of medical care and welfare.

The advantages of NIRS over fMRI and PET for evaluating brain function are as follows: (1) it is noninvasive, because it uses weak near-infrared light; (2) the equipment is compact and can be moved around, eliminating the need for special examination rooms; (3) the examinee does not need to be fixed to a bed and can be assessed in a natural state [18]. In general, the evaluation of brain function by NIRS involves capturing the brain's response when images or sound stimuli are given as cognitive tasks; however, few studies have presented these stimuli to humans rather than animals. NIRS brain function measurement is an excellent method to capture the mental and physiological changes caused by animal interventions, considering the specifications and advantages of the device. Previous authors focused on the prefrontal cortex when measuring brain function. The prefrontal cortex integrates and organizes stimuli from inside and outside the body through sensory organs and issues commands. The medial prefrontal cortex is the foundation of intellect and the mind. It is responsible for higher mental functions, such as planning, future prediction, emotional inhibition, working memory, motivation, thinking, interaction with others and self, and consciousness. Watanabe et al. reported that the medial prefrontal cortex was hypoactive in patients with autism spectrum disorders, affecting the ability to understand

others and communicate [19]. Capturing the activity of this brain region may be extremely important in determining the causes of the effects of animals on the human psyche and the causes of changes in the human psyche.

This study aimed to incorporate this into employment support activities at welfare facilities in cooperation with the Ishikawa Prefecture Disability Welfare Division and one of the welfare service offices for persons with disabilities. This study aimed to examine the effects of a one-day sheep-rearing experience on motivation and anxiety in patients with chronic psychiatric disorders. In particular, we focused on oxytocin and cortisol levels, apathy scales, and brain activity in the prefrontal cortex region, all of which have recently attracted attention as bioactive substances related to happiness and attachment, as well as the development of negative symptoms and decreased motivation that prevent patients with schizophrenia from returning to society and working.

## 2. Materials and Methods

### 2.1. Research Methods

#### 2.1.1. Participants

The participants were patients with psychiatric disorders requiring medical care who were attending psychiatric daycare (those diagnosed with schizophrenia, developmental disorders, bipolar disorders, anxiety disorders, or mood disorders according to DSM-V [20], the American Psychiatric Association diagnostic classification of psychiatric disorders). Eighteen participants were eligible for employment support services under the Comprehensive Support for Persons with Disabilities Act and were between 20 and 75 years old. Written informed consent was obtained from all participants, and they had written decision-making capacity.

#### 2.1.2. Study Design

This study involved a non-randomized, controlled trial. An intervention day with a sheep-rearing experience and a control day in the usual psychiatric daycare were set up at each facility in a crossover design with the cooperation of two facilities, A and B.

#### 2.1.3. Intervention

The participants' responses were compared to sheep-rearing experiential learning (1 day; intervention day) and regular daycare (1 day; control day) programs. On both days, the work hours were from 1:30 PM to 3:00 PM, and surveys were conducted from 1:00 PM to 1:30 PM (before the program) and from 3:00 PM to 3:30 PM (after the program).

Cortisol and oxytocin levels in saliva increase during sleep, are highest in the early morning, and decrease during daytime activity [21,22]. Therefore, the study was conducted in the afternoon, when the diurnal variation was considered, and fluctuations were minimal. A washout period of 2 months was established between the intervention and control days to minimize the effects of the intervention (Figure 1). The intervention day consisted mainly of touching and feeding the sheep. The control days were spent at the daycare center of the psychiatric hospital, where the participants usually visited, and comprised tasks (walking, mahjong, paper crafts, and boccia) they chose according to their mood (Figure 2). In each case, an employment support staff member was assigned to the participant and oversaw the work, ensuring that the workload was not excessive, that the workload was even according to the participant, and that the participant's physical condition was monitored on that day (Figure 1).

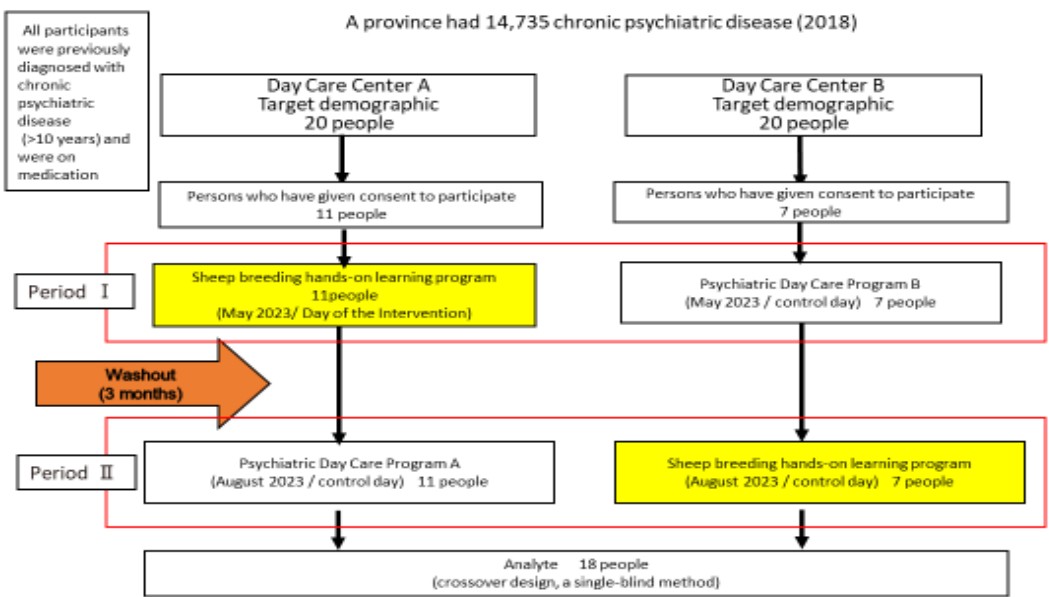

**Figure 1.** Flow of participants through each stage of the study, crossover design, single-blinded method.

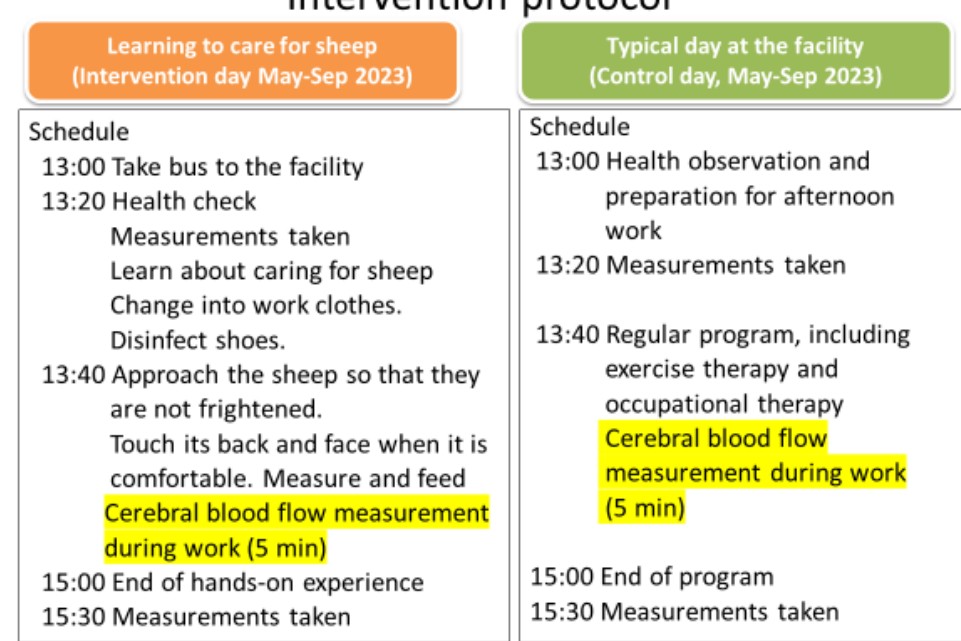

**Figure 2.** Intervention protocols. Sheep rearing experience day and psychiatric day care occupational therapy day.

A total of 40 potential participants were identified (20 in each of the 2 facilities), and 18 expressed interest in the study. The required sample size was 37 participants with an effect size of 0.8 (G*power3.1). Since this study included 18 participants, the effect size was calculated at 0.3, which was rather small.

*2.2. Survey Methodology*

2.2.1. Measurement of the Motivation Score (Apathy Scale)

The motivation score (apathy scale), a Japanese version of the apathy scale developed by Starkstein et al. [23] with verified reliability and validity [24], was used to assess

participants' motivation. Based on the results of the responses, we evaluated aspects of the participants' "interest", "concern", "concentration", "plans for the future", "emotions (facial expressions)", and "motivation" toward employment. The 14 questions were as follows (Table 1).

**Table 1.** Apathy scale.

| No. | Apathy Scale Questions |
| --- | --- |
| 1 | Are you interested in learning new things? |
| 2 | Do you have any interests? |
| 3 | Are you concerned about your condition? |
| 4 | Do you put much effort into things? |
| 5 | Are you always looking for something to do? |
| 6 | Do you have plans or goals for the future? |
| 7 | Do you have motivation? |
| 8 | Do you have the energy for daily activities? |
| 9 | Does someone have to tell you what to do each day? |
| 10 | Are you indifferent to everything? |
| 11 | Are you unconcerned with many things? |
| 12 | Do you need a push to get started on things? |
| 13 | Are you neither happy nor sad, just in between? |
| 14 | Would you consider yourself apathetic? |

### 2.2.2. Measurement of Changes in Salivary Cortisol and Oxytocin

The participants were briefed on the saliva collection process [25,26]. The saliva samples were then collected. The participants were prohibited from smoking, eating, or exercising for 30 min before collection, and a supervisor was present during collection to ensure that precautions were taken. Saliva was collected from each participant via a straw in a 1.5 mL polypropylene tube to a total volume of approximately 2 mL after rinsing the mouth lightly with water before collection. Simultaneously, a motivation questionnaire survey was self-administered. In addition, none of the participating women were menstruating.

### 2.2.3. Measurement of Cerebral Blood Flow Using NIRS

This study used the change in oxy-Hb as an evaluation criterion for brain activity [27–29]. Regarding NIRS, the participant's head was covered with a special holder, the optical fiber probe was fixed such that it was in close contact with the scalp, and near-infrared light was transmitted. NIRS is safe because it uses near-infrared light, which is harmless to the human body and has few restrictions on body movements. In this study, we measured and analyzed brain activation responses using a wearable multichannel NIRS system (cerebral blood flow) [Brite MK II] (Artinis).

### 2.2.4. Survey Period

The study period was from April 2022 to December 2023. The comparison of changes before and after the period of sheep rearing (intervention day) and normal occupational therapy (control day) during psychiatric daycare attendance was conducted with two groups of 18 patients who agreed to participate in the study. One group was involved in sheep-rearing (intervention day), and the other continued to perform the agricultural work they had been doing (control day). On both days, the questionnaire survey and saliva sampling were conducted at two time points: before and immediately after the intervention. In addition, frontal lobe blood flow was measured during 5 min of work on both days.

The two collaborating institutions were used in a crossover design, with Institution A starting on the control day and moving to the intervention day. Institution B started on the intervention day and moved to the control day. The changes in each item at the two time points on each measurement day were compared and validated (Figure 1).

*2.3. Analysis*

2.3.1. Identification of the Channel Location Using Standard Brain Coordinates

Even without MRI images of the participants, the channel position can be registered to the MNI standard brain coordinate system based on the International 10–20 method (using the established registration method) [30]. In this study, we used a holder created by Artinis and a 3D digitizer to position the probe holder such that a light-transmitting probe 7 (T7) was in the frontal pole FPZ. The channel position was then changed to the standard brain coordinate system and projected to the brain surface by the brain function mapping NIRS/analysis software "Oxysoft3.3.34.1 x64 (Artinis)" (Figure 3).

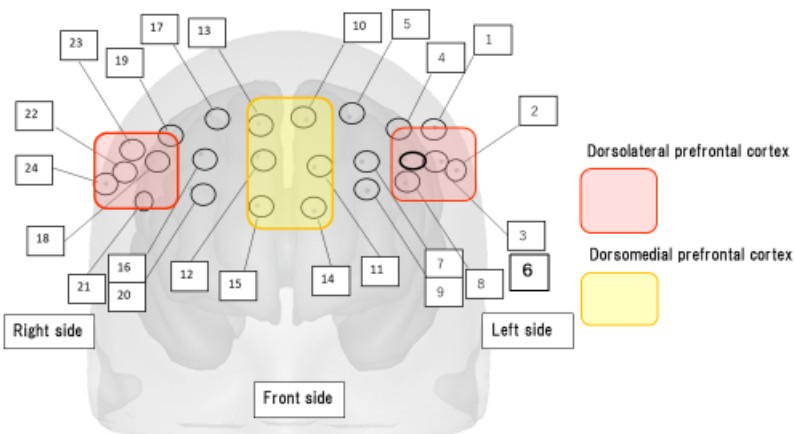

**Figure 3.** Brain coordinates and measurement positions at a total of 24 channel locations.

2.3.2. Comparison of Baseline Data on the Intervention and Control Days

First, salivary cortisol and oxytocin measurements and motivation score measurements were compared using the Mann–Whitney U test on the intervention and control days before the intervention began. Prefrontal blood flow test values were compared to 5 min integral oxygen hemoglobin concentrations (oxyHb) (for all 24 channels) during work on the intervention and control days before the start of the intervention, using the Mann–Whitney U test for differences between the groups at baseline.

2.3.3. Changes in the Intervention and Control Dates before and Immediately after the Start of the Intervention (Between-Group Comparisons)

The Wilcoxon signed-rank test was performed on salivary cortisol and oxytocin measurements and motivation score measurements before and immediately after the intervention on the intervention and control days. Differences between the intervention and control days were calculated and compared using the Mann–Whitney U test. Prefrontal blood flow test values were compared using the Mann–Whitney U test and the oxy-Hb integral values (for all 24 channels) for 5 min during work on the intervention day and the control day. IBM SPSS 25 for Windows was used for all statistical analyses, with a significance level of 5%.

### 2.4. Ethical Considerations

The study's purpose, outline, and ethical considerations were explained in writing and orally to the heads of the cooperating institutions, and their consent to participate in the study was obtained. After that, the purpose of the research was explained to the users and employment support staff orally and in the form of an explanation sheet. We explained the study outline, assuring them that they would not suffer any disadvantage by not participating, that anonymity would be ensured so that individuals and facility names could not be identified, and that the data would be strictly managed. This study was conducted with the approval of the Ethics Committee for Human Subjects of the university to which the researcher belonged (Nursing No. R4-14). The trial registration number is UMIN000050224.

### 3. Results

#### 3.1. Participants' Characteristics

Among the eighteen participants (50.94 ± 14.52 years old; seven males) who consented to participation in the study, eleven were from Institution A (52.30 ± 15.68 years old; four males), and seven were from Institution B (49.00 ± 12.43 years old; two males) (Table 2). The psychiatric disorders present were schizophrenia in eight, autism spectrum disorder in four, bipolar disorder in four, and mood disorder in two. The age, cortisol, oxytocin levels, and motivation scores were compared between the groups at sites A and B before the intervention began (at baseline) (Table 3). These comparisons revealed no significant differences between the two groups.

**Table 2.** Participant characteristics.

|    | Facility | Sex | Age | Disease | Medication |
|----|----------|--------|-----|----------------------------------|---------------------------------------------|
| 1  | A | Male | 75 | Schizophrenia | Risperidone |
| 2  | A | Female | 51 | Schizophrenia | Risperidone |
| 3  | A | Female | 34 | Extensive Developmental Disorder | Bronanserin, Sulpiride |
| 4  | A | Female | 37 | Schizophrenia | Zepreon, Abilify |
| 5  | A | Male | 50 | Schizophrenia | Aripiprazole |
| 6  | A | Male | 51 | Bipolar Mood Disorder | Risperidone, Lithium |
| 7  | A | Female | 73 | Schizophrenia | Risperidone |
| 8  | A | Female | 73 | Schizophrenia | Blonanserin |
| 9  | A | Female | 34 | Extensive Developmental Disorder | Bronanserin, Sulpiride |
| 10 | A | Male | 30 | Developmental Disorder | Strattera |
| 11 | A | Female | 49 | Schizophrenia | Risperidone, Serenace, Quetiapine |
| 12 | B | Male | 65 | Bipolar Disorder | Tegretol, Duloxetine capsules, Tsumura |
| 13 | B | Female | 49 | Bipolar Disorder | Olanzapine, Abilify |
| 14 | B | Female | 45 | Schizophrenia | Sobotril, Abilify |
| 15 | B | Female | 31 | Anxiety Disorders, Bipolar Disorder | Iron tablets, vitamins, zinc |
| 16 | B | Female | 34 | Autism Spectrum | Invega, Tegrelor, Lorazepam |
| 17 | B | Female | 64 | Depression | Sertraline, Abilify |
| 18 | B | male | 55 | Depression | Quetiapine, Rhythmic, Sodium valproate tablets |

**Table 3.** Comparison at baseline (intervention and control days).

| | | Facility A n = 11 | | Facility B n = 7 | | *p* Value *1 |
|---|---|---|---|---|---|---|
| | Unit | Mean Value | Standard Deviation | Mean Value | Standard Deviation | |
| Age (years) | | 50.64 | 16.62 | 49.00 | 13.43 | 0.785 |
| Pre-intervention Oxytocin | (μg/dL) | 0.12 | 0.07 | 0.15 | 0.07 | 0.495 |
| Pre-intervention Cortisol | (pg/mL) | 315.88 | 252.72 | 276.78 | 349.55 | 0.051 |
| Pre-intervention Apathy Scale Score | | 25.36 | 8.64 | 28.00 | 11.39 | 0.554 |
| Pre-control Oxytocin | (μg/dL) | 0.16 | 0.11 | 0.18 | 0.07 | 0.315 |
| Pre-control Cortisol | (pg/mL) | 337.44 | 270.99 | 425.10 | 407.23 | 0.891 |
| Pre-control Apathy Scale Score | | 25.60 | 8.45 | 26.86 | 7.82 | 0.768 |

*1: Mann–Whitney U test.

*3.2. Comparison of the Salivary Oxytocin and Cortisol Levels before and after the Intervention*

A Wilcoxon signed-rank test was conducted on the motivation scores and salivary cortisol and oxytocin levels before and immediately after the intervention on the intervention and control days (Table 4). The results showed that motivation scores were significantly higher after the intervention than before on both the intervention and control days. In contrast, the oxytocin levels did not change significantly on either day; however, the oxytocin levels on the intervention day had a higher mean value after the intervention ($p$ = 0.062). The cortisol levels did not change significantly on either day; however, the cortisol levels on the control day had a lower mean value after the intervention ($p$ = 0.055). The Wilcoxon signed-rank test of the changes in motivation score and oxytocin and cortisol levels on the intervention and control days revealed no significant changes in any of the values (Table 5).

**Table 4.** Comparison of apathy scale scores and oxytocin and cortisol levels before and after the intervention n = 18.

| | | Before Intervention | | After Intervention | | *p* Value *1 |
|---|---|---|---|---|---|---|
| Item | | Mean Value | Standard Deviation | Mean Value | Standard Deviation | |
| Intervention Day Apathy Scale Score | | 26.389 | 9.562 | 29.167 | 8.952 | 0.003 |
| Intervention Day Cortisol | (μg/dL) | 0.133 | 0.070 | 0.129 | 0.140 | 0.435 |
| Intervention Day Oxytocin | (pg/mL) | 299.781 | 286.596 | 368.258 | 385.218 | 0.062 |
| Control Day Apathy Scale Score | | 26.118 | 7.968 | 30.000 | 7.802 | 0.003 |
| Control Day Cortisol | (μg/dL) | 0.164 | 0.097 | 0.126 | 0.093 | 0.055 |
| Control Day Oxytocin | (pg/mL) | 371.531 | 321.965 | 356.488 | 416.676 | 0.523 |

*1: Wilcoxon signed-rank test.

**Table 5.** Comparison of the changes in apathy scale scores and oxytocin and cortisol levels between the intervention and control days, n = 18.

| | Unit | Intervention Day | | Control Day | | *p* Value *1 |
|---|---|---|---|---|---|---|
| | | Mean Value | Standard Deviation | Mean Value | Standard Deviation | |
| Apathy Scale Score change | | 2.778 | 3.439 | 3.882 | 4.498 | 0.531 |
| Cortisol change | (μg/dL) | 0.001 | 0.103 | −0.039 | 0.097 | 0.379 |
| Oxytocin change | (pg/mL) | 80.475 | 155.626 | −15.043 | 265.990 | 0.093 |

*1: Wilcoxon signed-rank test.

### 3.3. Test for Carryover Effect

To examine the carryover effect of the crossover design, the Mann–Whitney U test was performed on Sequences 1 and 2 based on Table 6. The results were as follows: apathy scale $p = 1.000$, cortisol $p = 0.751$, oxytocin $p = 0.821$. There was no effect of carryover effect on any of the values.

**Table 6.** Carryover and treatment effects of crossover design.

| Subject | Sequence | Carryover Effect | | | Treatment Effect | | |
|---|---|---|---|---|---|---|---|
| | | Apathy Scale | Cortisol Change | Oxytocin Change | Apathy Scale | Cortisol Change | Oxytocin Change |
| 1 | 1 (AB) | 12 | −0.24 | −760.85 | 2 | 0.04 | −500.45 |
| 2 | 1 (AB) | 10 | 0.14 | −41.25 | 0 | 0.18 | −53.25 |
| 3 | 1 (AB) | 1 | −0.1 | 28.81 | 1 | 0 | 106.95 |
| 4 | 1 (AB) | 4 | −0.01 | −84.6 | 0 | −0.03 | 89.38 |
| 5 | 1 (AB) | 2 | 0.09 | −63.58 | −2 | −0.01 | 63.58 |
| 6 | 1 (AB) | 0 | −0.04 | 186.45 | 2 | −0.04 | −291.37 |
| 7 | 1 (AB) | 3 | 0.02 | 171.67 | −1 | 0 | −102.47 |
| 8 | 1 (AB) | 6 | −0.21 | 69.26 | 4 | −0.25 | −146.8 |
| 9 | 1 (AB) | 5 | −0.21 | 49.89 | −5 | −0.21 | 49.89 |
| 10 | 1 (AB) | 8 | −0.06 | 102.66 | 8 | −0.04 | −203.12 |
| 11 | 1 (AB) | 12 | 0.1 | 51.35 | −4 | −0.2 | −58.55 |
| 12 | 2 (BA) | 1 | −0.11 | 12.1 | 1 | −0.03 | −32.08 |
| 13 | 2 (BA) | 2 | −0.07 | −8.81 | 4 | −0.07 | 17.09 |
| 14 | 2 (BA) | 18 | 0.03 | 699.79 | −14 | 0.03 | −35.21 |
| 15 | 2 (BA) | 6 | −0.12 | −162.45 | −6 | 0.02 | 223.59 |
| 16 | 2 (BA) | 21 | −0.12 | 16.64 | 3 | 0.00 | 84.00 |
| 17 | 2 (BA) | 3 | −0.17 | −313.41 | 7 | −0.01 | 522.65 |
| 18 | 2 (BA) | 2 | 0.45 | 1143.67 | −6 | 0.17 | −187.41 |

### 3.4. Changes in Cerebral Blood Flow on the Intervention and Control Days Using NIRS

The Mann–Whitney U test was used to compare the changes in cerebral blood flow activity (oxy-Hb) during the sheep-rearing experience (5 min) and participation in the regular psychiatric daycare program (5 min) of the participants (15 participants). The

results showed that the amount of change was different for each channel. Although the amount of change differed from channel to channel, significant differences were found in channels 4 and 16 on the intervention day, showing a statistically significant increase in brain activity (Table 7, Figure 4). In addition, a comparison of individual changes in cerebral blood flow activity in two males and two females with autism spectrum disorder showed that the left dorsolateral prefrontal region was active on the intervention day (Figures 5–8).

**Table 7.** Changes in cerebral blood flow (measured in 15 participants) during the sheep-rearing experience (intervention) and usual occupational therapy (control) were compared (Mann–Whitney U test), n = 15.

| Brain Coordinate Measurement Position | | Oxygen Hemoglobin Concentration (Integral) | Mann–Whitney U *p*-Value |
|---|---|---|---|
| Channel 1 | Intervention | −57,010.491 | 0.458 |
| | Control | −28,992.879 | |
| Channel 2 | Intervention | −29,273.356 | 0.652 |
| | Control | −8469.174 | |
| Channel 3 | Intervention | −14,776.647 | 0.852 |
| | Control | −18,856.741 | |
| Channel 4 | Intervention | 30,126.428 | 0.032 |
| | Control | −58,048.605 | |
| Channel 5 | Intervention | −2770.151 | 0.942 |
| | Control | 1090.629 | |
| Channel 6 | Intervention | 15,648.397 | 0.270 |
| | Control | −8267.619 | |
| Channel 7 | Intervention | 6226.593 | 0.553 |
| | Control | −19,568.200 | |
| Channel 8 | Intervention | −27,745.876 | 0.691 |
| | Control | −5306.442 | |
| Channel 9 | Intervention | −83,291.332 | 0.151 |
| | Control | −8779.836 | |
| Channel 10 | Intervention | −53,775.030 | 0.568 |
| | Control | −23,492.277 | |
| Channel 11 | Intervention | 37,053.522 | 0.452 |
| | Control | 14,198.655 | |
| Channel 12 | Intervention | −51,526.818 | 0.280 |
| | Control | 2286.230 | |
| Channel 13 | Intervention | 9789.809 | 0.808 |
| | Control | 1629.656 | |
| Channel 14 | Intervention | 18,595.649 | 0.978 |
| | Control | 17,685.441 | |
| Channel 15 | Intervention | −1839.110 | 0.443 |
| | Control | −18,314.892 | |

**Table 7.** *Cont.*

| Brain Coordinate Measurement Position | | Oxygen Hemoglobin Concentration (Integral) | Mann–Whitney U *p*-Value |
|---|---|---|---|
| Channel 16 | Intervention | 33,583.087 | 0.045 |
| | Control | −20,404.002 | |
| Channel 17 | Intervention | 117.952 | 0.733 |
| | Control | −17,721.914 | |
| Channel 18 | Intervention | 3192.267 | 0.577 |
| | Control | −8242.235 | |
| Channel 19 | Intervention | −3593.690 | 0.715 |
| | Control | −8797.640 | |
| Channel 20 | Intervention | 16,986.230 | 0.330 |
| | Control | 1771.079 | |
| Channel 21 | Intervention | −21,777.784 | 0.467 |
| | Control | 787.291 | |
| Channel 22 | Intervention | −25,666.015 | 0.745 |
| | Control | −12,234.070 | |
| Channel 23 | Intervention | 31,754.014 | 0.098 |
| | Control | −13,519.307 | |
| Channel 24 | Intervention | 17,887.825 | 0.645 |
| | Control | −170.636 | |

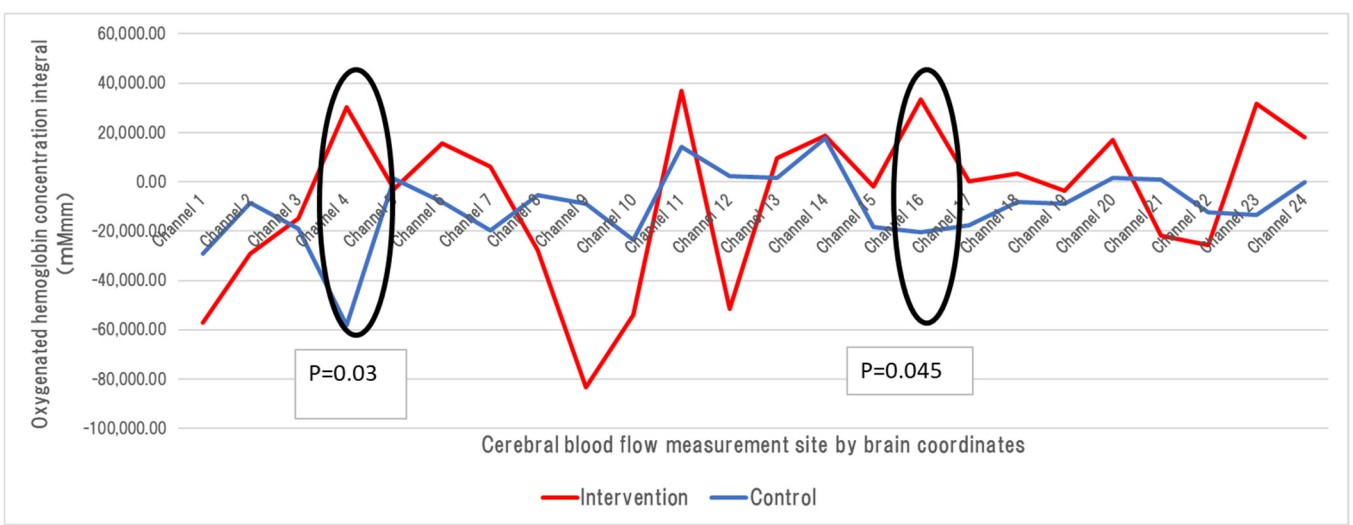

**Figure 4.** Changes in cerebral blood flow—comparison between sheep rearing and occupational therapy (5 min measurement).

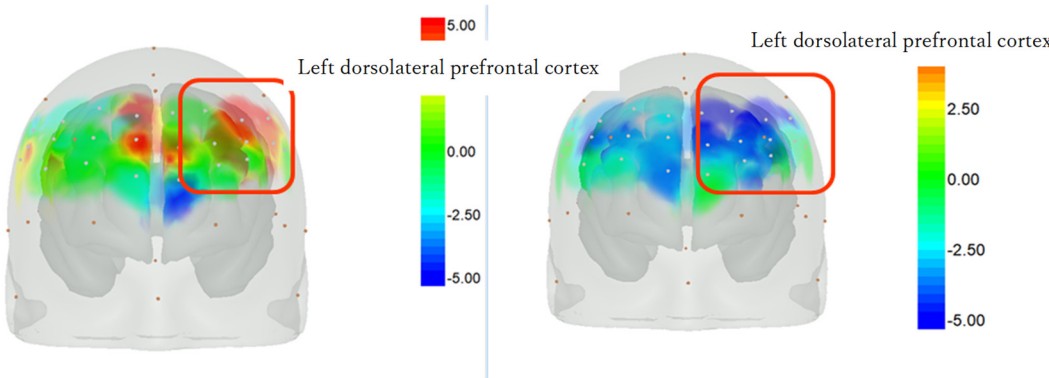

**Figure 5.** (**Left**) During the sheep-rearing experience. Changes in cerebral blood flow (5 min during intervention) for Female No. 16 with autism spectrum disorder. (**Right**) Changes in cerebral blood flow during occupational therapy (Poccia) (5 min during intervention) for Female No. 16 with autism spectrum disorder.

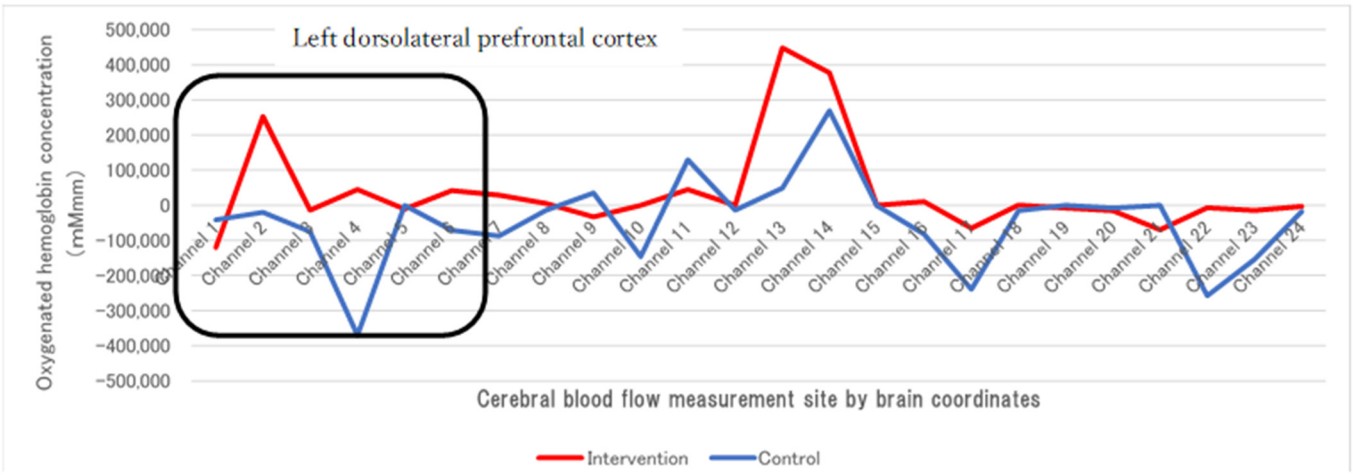

**Figure 6.** Changes in cerebral blood flow—comparison between sheep rearing and occupational therapy (5 min measurement).

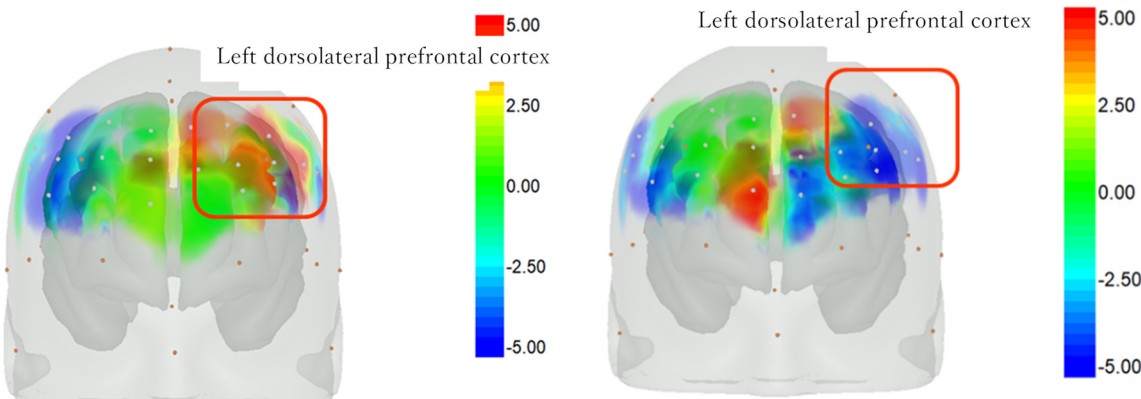

**Figure 7.** (**Left**) During the sheep-rearing experience. Changes in cerebral blood flow (5 min during intervention) for Male No. 10 with a developmental disability. (**Right**) Changes in cerebral blood flow during occupational therapy (Poccia) (5 min, during intervention) for Male No. 10 with a developmental disability.

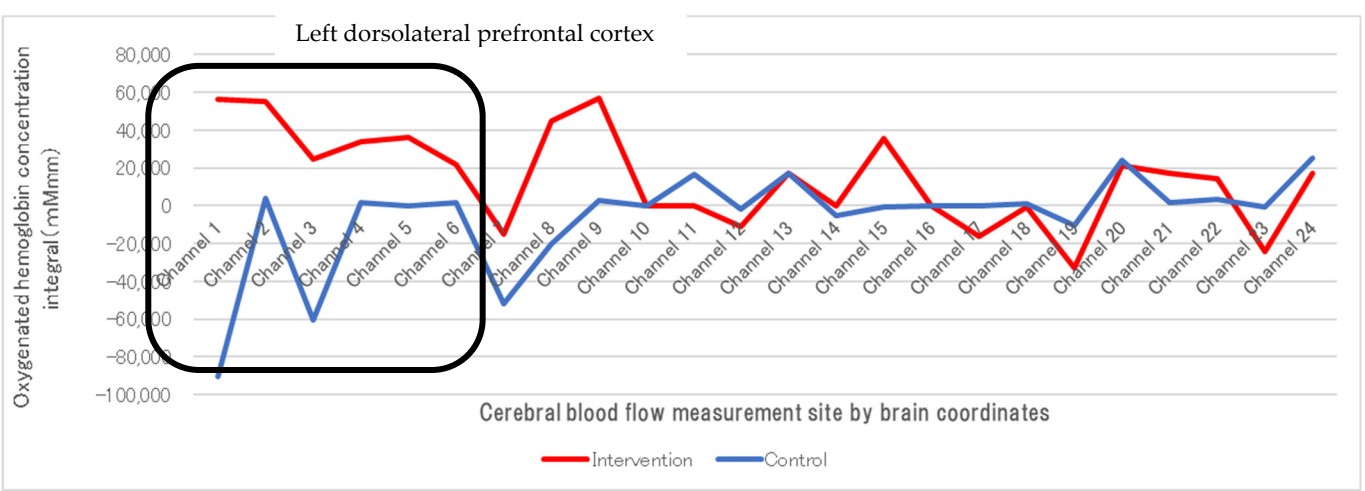

**Figure 8.** Changes in cerebral blood flow during sheep-breeding experience and usual occupational therapy (Poccia) (integral values).

## 4. Discussion

### 4.1. Participants' Characteristics

All the study participants had psychiatric diseases and were taking medications. The highest number of participants with schizophrenia lived in the community while taking their medications; however, it was difficult for them to return to society (i.e., return to work). Previous studies on people with psychiatric diseases and job retention have shown that schizophrenia, depression, sleep disorders, the duration of illness (long-term), age (younger age group), attention disorder, social cognitive disorder, full-time employment, lack of work-related counselors, long daily working hours, short duration of employment, and other factors may increase the risk of leaving employment [3–5]. The results also indicated that the participants did not receive training and support before and after entering the workforce and did not feel fulfilled in their jobs. The apathy scale result at baseline for the current participants was 19 points. If 16 points or more is taken as a state of low motivation [23,24], it can be concluded that the current participants were in a state of low motivation.

### 4.2. Changes in the Apathy Scale Scores during the Sheep-Rearing Experience and Usual Occupational Therapy

The apathy scale scores of the participants showed statistically significant improvement on the intervention and control days. The psychiatric daycare program is a part of the medical care for persons with psychiatric disorders who are in the process of recovery. Furthermore, its purpose is to provide them with lifestyle and work guidance under appropriate medical supervision to facilitate their smooth reintegration into society (Ministry of Health, Labor and Welfare 2009). This program aims to improve the motivation to return to society through participation. The fact that the intervention effect of the sheep-rearing experience in this study was the same as that of the daycare program at a medical institution indicates that this program can also be used as part of a psychiatric rehabilitation program.

Recently, AATs have been widely used in medical and welfare facilities. AAT is used to improve the functioning of people with dysfunctions or disabilities because of illness or other causes. When people with disabilities interact with animals, physical, mental, and social benefits (such as emotional stability, recreation, and improvement in quality of life) can be expected. Moreover, AAT is attracting attention as a complementary and alternative therapy [11–17].

This study also focused on agricultural work as an occupation. Our results can help improve the motivation to work for people with psychiatric disabilities, whose motivation tends to decline, through intervention using animals and agricultural work. In a previous

study conducted by Shimizu et al. [31], people with psychiatric disabilities who had experienced sheep rearing showed an increase in testosterone levels compared with those who received usual occupational therapy. This finding indicates that sheep-rearing has an impact on improving the motivation of people with psychiatric disabilities.

### 4.3. Changes in Salivary Oxytocin and Cortisol Levels during the Sheep-Rearing Experience and Usual Occupational Therapy

Regarding the changes in oxytocin and cortisol levels as physiological indicators between the intervention and control days, there was no statistically significant difference regarding oxytocin on the intervention day; however, the mean value after the intervention showed an improvement compared with that before the intervention (Table 3). Furthermore, when comparing the pre- and post-intervention changes in cortisol and oxytocin levels on the intervention and control days (Table 4), the mean oxytocin change was +80 (pg/mL) on the intervention day. However, on the control day, the mean oxytocin change was −15 (pg/mL) which is a negative value but not statistically significant.

Oxytocin is a peptide hormone comprising nine amino acids secreted by the posterior pituitary gland. Henry Dale, in 1906, found that a pituitary extract had uterine contraction effects [32]. This effect was also observed by Vigneaud et al. in 1953 [33]. It is now believed that oxytocin has central and psychoactive effects in addition to its effects on peripheral tissues, such as uterine contraction, and it is involved in social behavior (communication, including reproductive and aggressive behaviors) and the establishment of trusting relationships [34,35].

Analyses in humans have reported that oxytocin improves the ability to infer the psychology of others and perceive and recognize emotions based on their facial expressions [36–38]. Based on these reports, oxytocin is expected to be useful in improving psychiatric disease-like symptoms. Oxytocin has been verified for its therapeutic efficacy in autism, and improvements in the response to the facial expressions of others have been reported [39]. In addition to reports that patients with frontotemporal dementia who receive oxytocin are less likely to perceive anger or fear in others' facial expressions, its efficacy in patients with PTSD has also been examined [40,41]. Oxytocin is an important hormone that facilitates social life in patients with psychiatric disorders. Shamay-Tsoory and Abu-Akel reported the following major psychoactive effects of oxytocin: (1) the promotion of social skills; (2) reduction in fear and stress levels; (3) the enhancement of trust within an attributed organization; (4) reduction in anxiety levels; (5) the enhancement of altruistic behavior [42,43]. This study showed that oxytocin levels tended to increase after the sheep-rearing intervention compared with participation in a regular psychiatric daycare program. These results indicate that incorporating animal rearing as part of employment support or the continuous experience of rearing livestock as part of a psychiatric daycare program that includes employment training may help increase oxytocin levels. This increase can improve the ability of psychiatric patients to infer the psychology of others, improve their ability to perceive and recognize emotions based on the facial expressions of others, and lead to better relationships with others.

There was a decreasing trend for the cortisol levels after the intervention on the control day compared to the intervention day. However, the difference was not significant. Cortisol is a steroid hormone released from the adrenal cortex and is most often studied with respect to stress. Cortisol has various physiological effects on the immune, vascular, and central nervous systems and is an important hormone for psychological and physical health [44]. In addition, saliva cortisol correlates well with blood cortisol [45] and can be collected non-invasively.

The fact that the decrease in cortisol levels was more pronounced during the regular psychiatric daycare program than during the sheep-rearing experience demonstrates that psychiatric patients, who tend to be more anxious than healthy people, may have been more anxious when participating in a program at their usual place of habitual attendance. The cortisol levels did not increase on the intervention day; however, the fact that the

levels did not increase demonstrates that the sheep-rearing experience did not increase anxiety levels.

In a previous study, the salivary cortisol levels of students who participated in a dog interaction activity decreased significantly after participation [45]. These results demonstrated that participation in animal interaction activities may reduce stress in students. In addition, although the effects of stress reduction have been reported by measuring salivary amylase levels in people with dementia, there are no reports on the effects of animal-mediated activities using physiological indicators in patients with schizophrenia, developmental disorders, or other psychiatric disorders.

Salivary cortisol has attracted attention as a physiological assessment of stress levels [46]. Cortisol has various physiological effects on the immune, vascular, and central nervous systems and is an essential hormone in psychological and physical health. In particular, a previous study by Shimizu et al. [31] reported that salivary cortisol levels significantly decreased after sheep rearing and were strongly associated with anxiety in patients with psychiatric disorders. Although there was no significant decrease on the day of the intervention, the results indicated that the sheep-rearing experience did not increase anxiety in patients with psychiatric disorders, who tend to be more anxious and vulnerable to stress when introduced to new experiences.

### 4.4. Changes in Brain Activity during the Sheep-Rearing Experience and Usual Occupational Therapy

Continuous measurements of cerebral blood flow during 5 min of work on the intervention and control days, among which the oxy-Hb integral values were compared for each of the 24 channels, showed a significant activation of the values on the intervention day compared to the control day in the two left prefrontal regions of channels 4 and 16 (Table 4, Figure 4). Thus, all participants in this study exhibited significant cerebral blood flow activity because of participating in the sheep-rearing program. On the control day, when the participants participated in the regular daycare program, the program included static tasks (e.g., paper crafts and mahjong) and dynamic tasks, such as Boccia competitions. Therefore, it cannot be said that the significantly greater activity during the intervention day than during the control day was because the sheep-rearing experience was dynamic.

Channel 4, the prefrontal cortex region that showed activity in this study, was likely the left dorsolateral prefrontal cortex (DLPFC) [47]. Many studies have proposed that the left DLPFC is activated during verbal working memory tasks, such as attention retention, thinking, judgment, and problem solving [48]. The DLPFC is responsible for executive functions during task performance, including conscious attentional control, the efficient performance of working memory, attention allocation, and attention translation tasks [47]. The dysfunction of the DLPFC is one of the most reproducible findings in schizophrenia research, especially in tasks requiring cognitive control [49–55]. Yoon, J. H. et al. found that in univariate contrasts between schizophrenic patients and healthy controls, only the dorsal prefrontal cortex (dorsal prefrontal cortex) was associated with cognitive control deficits and related behavioral abnormalities in schizophrenia [54]. They found differences in activity only in the DLPFC (but not in other elements of the task-related neural network) in univariate contrasts between schizophrenic patients and healthy controls, and in a logistic regression analysis, they reported that reduced activity in the DLPFC was diagnostic of schizophrenia. They further concluded that the impairment of the DLPFC in schizophrenic patients was basic and unrelated to the long-term effects of medication or chronicity of the illness Weinberger et al. 1986 [55]. Carter et al. also reported that prefrontal hypofunction is a marker of PFC dysfunction in schizophrenia and most reliably occurs during cognitive tasks that load on PFC function [55,56]. Furthermore, Joseph et al. stated that neuronal pathology in the dorsal PFC is likely the cause of abnormal physiological responses across the cortex in working memory.

Why did the sheep-raising intervention lead to DLPFC activity in schizophrenics with such reduced activity in the DLPFC? Previous AAA studies have not investigated animal intervention and DLPFC activity.

However, in a previous study, Kramer et al. found that 6 months of aerobic exercise in older adults increased DLPFC activity during executive function tasks [56]. Moreover, a meta-analysis of studies examining the effects of exercise on cognitive function showed that the prefrontal cortex, including the DLPFC, is an area that is responsive to exercise [57]. These findings may explain why exercise contributed to the activity of the DLPFC in the present study. Both tasks in this study involved physical activity. The major difference was the presence or absence of an animal intervention, such as "sheep-rearing", which may have had an impact on the patients with a psychiatric disability that has not been observed in psychiatric occupational therapy to date.

### 4.5. Bilateral Frontal Lobe Activation with and without an Animal Intervention

This study compared prefrontal cortex activation by two intervention methods, sheep rearing and psychiatric occupational therapy, which showed that the sheep-rearing day tended to activate the DLPFC for 5 min during work and improved oxytocin levels (Table 4, Figure 4).

Although there was an increase in motivation, as measured by the apathy scale, on both days, increased brain activity and a trend toward an increase in oxytocin levels were observed only on the intervention day. This change in brain activity and the trend toward increased oxytocin levels may have been because of the presence or absence of animal intervention. The intervention day marked the participants' first experience interacting with and feeding sheep, and they were most uncomfortable with the animals. However, through the sheep-rearing experience, the participants became emotionally involved with the sheep by gazing at their faces and interacting with the sheep that rubbed up against people, which may have stimulated the activation of their DLPFC more than occupational therapy.

In the temporal and frontal lobes of adults with autistic disorder, it has been reported that there is a lower decrease in cortical volume and thickness with age in healthy participants [58]. However, the gray matter volume is usually smaller in the insula, lower frontal lobe, and lower parietal lobe than in healthy participants of the same age [59]. Other methods that examine axonal projections, such as diffuse tensor imaging, have reported reduced fractional anisotropy values in the medial prefrontal cortex, anterior cingulate gyrus, limbic system, and white matter at the temporal–parietal lobe boundary [60].

Deeley et al. [61] reported that their autism group exhibited reduced activation in the spindle gyrus and visual association cortex in their expressions of joy, anger, and sadness compared to the normal group. However, there was no difference in neutral expressions. The activation differed between the faces of strangers and family members, with the autistic group showing a reduction in the spindle gyrus in the former. In contrast, Dapretto et al. [62] reported the reduced activation of frontal mirror neurons in areas 44 and 45 in their adolescent autistic disorder group, which had no abnormalities in facial expression discrimination. The F5 area, which contains mirror neurons, is in the premotor area of the frontal lobe, an integrative area just anterior to the primary motor cortex that controls elemental movements, integrated movements, and actions with a high degree of skillfulness in conjunction with information from visual and auditory sources. The sheep-rearing experience provided an opportunity to learn about the importance of vigilance and alertness. We believe that the sheep-rearing experience focused nerves on actions, such as closely observing the movements and characteristics of the alert and cautious sheep and fearfully touching them. Moreover, channel 16 may have been more active during sheep-rearing than during therapy at psychiatric daycare.

Previous studies on AAT have reported that conversation and social activity increased when an intervention using dogs was conducted with people living in a nursing home [63]. Moreover, when the therapeutic effects of speech therapy and AAT were confirmed in patients with aphasia because of left cerebral injury, the patients were more motivated to

attend therapy during AAT, and stress was reduced during sessions, which the interactions between animals and humans may have influenced [64]. A previous study reported that dog-mediated therapy is effective for verbal and non-verbal communication in patients with aphasia [45]. However, the fact that dog–human communication is useful for training verbal and emotional expression, which is an expression of higher brain functions, indicates that dogs are also social animals that can recognize human faces [65]. Similarly, breeding sheep that can recognize human faces can be incorporated into psychiatric daycare and continuous employment support programs, leading to non-verbal communication with people with psychiatric disorders and having a certain effect on frontal lobe functions.

### 4.6. Limitations and Future Challenges

The small number of participants in this study (18) and the fact that it was not a random sampling may have biased the results. To further generalize the results of this study, the number of participants should be increased, and the survey should be continued. Non-random sampling was conducted at the request of the participants under the guidance of the facility's support staff. Random sampling may become feasible in the future as the results of this study are publicized and the number of research collaborators increases. We would also like to publish the results of the study to clarify the mechanism by which animal intervention contributed to the activity of DLPFC in mentally ill patients and to widely invite cooperation from welfare facilities for the disabled and livestock farmers in order to further verify its effectiveness and put the intervention into practice.

### 5. Conclusions

This study examined the effects of a one-day sheep-rearing experience on motivation and anxiety in patients with chronic-phase schizophrenia. In particular, we focused on oxytocin and cortisol levels, which have been reported to be associated with the onset of negative symptoms and decreased motivation and cognitive function that hinder the reintegration of patients with schizophrenia into society and their motivation to work. We compared the effects of the sheep-rearing experience on patients with chronic-phase schizophrenia during work therapy with those during usual daycare activities. The results of the comparison revealed the following:

(1) The fact that the sheep-rearing experience in this study showed the same apathy scale effect as the daycare program at the medical institution indicates that this project can be used as part of a psychiatric rehabilitation program.

(2) Oxytocin levels increased more after the sheep-rearing experience than the regular psychiatric daycare programs. These findings indicate that including animal husbandry in employment support or the continuous experience of rearing livestock as part of psychiatric daycare programs, which also serves as employment training, may increase oxytocin levels. This increase may improve the ability of psychiatric patients to infer the psychology of others, improve their ability to perceive and recognize emotions based on the facial expressions of others and lead to better relationships with others.

**Author Contributions:** Conceptualization, N.S. and M.I.; methodology, N.S.; software, N.S. and S.O.; validation, M.I., K.A. and S.O.; formal analysis, N.S.; investigation, N.S. and S.O.; resources, M.I.; data curation, M.I.; writing—original draft preparation, N.S.; writing—review and editing, N.S.; visualization, N.S.; supervision, M.I.; project administration, M.I.; funding acquisition, M.I. All authors have read and agreed to the published version of the manuscript.

**Funding:** This research was supported by the research program on developing innovative technology grants from the Project of the Bio-oriented Technology Research Advancement Institution (BRAIN) Grant Number JPJ007097.

**Institutional Review Board Statement:** The study was conducted according to the guidelines of the Declaration of Helsinki and approved by the Ethics Committee for Human Subjects of the university to which the researcher belonged (KANGO No. R4-14, Approval Date, 22 September 2022).

**Informed Consent Statement:** The purpose, outline, and ethical considerations of the study were explained in writing and orally to the heads of cooperating institutions, and their consent to participate in the study was obtained. After that, the purpose of the research was explained to the users and employment support staff orally and in the form of an explanation sheet. We explained the study outline, assuring them that they would not suffer any disadvantage by not participating, that anonymity would be ensured so that individuals and facility names could not be identified, and that the data would be strictly managed. Informed consent was obtained from all participants involved in the study.

**Data Availability Statement:** Figure 1. Flow of participants through each stage of the study, crossover design, a single-blinded method 10.6084/m9.figshare.25201856, 20 February 2023. Figure 2. Intervention Protocol 10.6084/m9.figshare.2520188310.6084/m9.figshare.25201883, 20 February 2023. Figure 3. Brain coordinates and measurement positions at 24 channel locations 10.6084/m9.figshare.25201898, 20 February 2023. Figure 4. Changes in cerebral blood flow (measured in 15 participants) during the sheep-rearing experience (intervention) and usual occupational therapy (control) were compared (Mann–Whitney U test) 10.6084/m9.figshare.25201910, 20 February 2023.

**Acknowledgments:** We would like to express our deepest gratitude to all staff members and users of the Type B continuous employment support facilities and their families who cooperated in this project.

**Conflicts of Interest:** The authors declare no conflicts of interest.

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
