# Peer review of "Effects of a One-Day Experiential Sheep-Rearing Experience on Motivation, Anxiety, and Frontal Lobe Brain Activity in Patients with Chronic Psychiatric Disorders: A Crossover Pilot Study"

_2673-5318, doi:10.3390/psychiatryint5020010_

Round 1

Reviewer 1 Report

Comments and Suggestions for Authors

Hello:

Thanks for giving me an opportunity to review. Your study is very well designed and an interesting read. I can personally attest to the positive effects my dog has had on my anxiety. However, here are some tips to make the study stronger:

- As it is a crossover study, it needs to be taken into consideration while performing statistical analysis.

- See this link: https://www.ibm.com/support/pages/worked-example-simple-crossover-design

- https://www.youtube.com/watch?v=p2UkYmSleLI

Author Response

13 Mar 2024.

Prof. Dr. Paolo Girardi

Editor-in-Chief,

Psychiatry International

Dear Editor:

We wish to re-submit the manuscript titled “The Effects of a One-Day Experiential Sheep-Rearing Experience on Motivation, Anxiety, and Frontal Lobe Brain Activity in Patients with Chronic Mental Illness: A Crossover Pilot Study.” The manuscript ID is psychiatryint-2903054.

We thank you and the reviewers for your thoughtful suggestions and insights. The manuscript has benefited from these insightful suggestions. I look forward to working with you and the reviewers to move this manuscript closer to publication in Psychiatry International.

The manuscript has been rechecked and the necessary changes have been made in accordance with the reviewers’ suggestions. The responses to all comments have been prepared and attached herewith.

Thank you for your consideration. I look forward to hearing from you.

Sincerely,

Nobuko Shimizu

Faculty of Nursing, Toyama Prefectural University

2-2-78 Nishinagae, Toyama City, Toyama Prefecture

930-0975

Email: n-shimizu@pu-toyama.ac.jp

Tel: +81-76-464-5410

Reviewer 2 Report

Comments and Suggestions for Authors

This is a very interesting and well-written article. It is clear and well-organized. There is a dearth of literature on how to encourage motivation among patients with chronic severe mental illness, especially schizophrenia. It is so important to add to the literature regarding unique ways of reducing isolation in this population.

The Abstract is very good.

Those outside of Japan may not know what a "Cabinet Office" is and a brief statement would be useful.

In line #35, OECD should be spelled out.

The Introduction is very good, especially the brief, yet strong, description of schizophrenia. Also, the classification of animal assisted therapy and activities. A good brief overview of evaluations of brain function.

The Materials and Methods section is easy to understand. A good description of the participants, the study design, and the intervention itself. It is tempting to criticize the short amount of time participants spent with the sheep but apparently that was all that was needed!

Figures 1 and 2 are helpful to the reader in understanding the sample and intervention protocols.

Very good to use the Apathy Scale and to include the 14 questions. While not a big deal, the authors may want to place the 14 questions as boxed material and call it Figure 3.

I am not a methodologist/statistician and hope a reviewer with that expertise can be helpful here. The current Figure 3, the 24 channel locations is complex.

The authors nicely addressed the ethical considerations and Ethics Committee Approval.

In the Results section, it was very helpful to have Table 1 provide details of patient characteristics, including diagnoses and medications. The figures showing changes in cerebral blood flow are especially interesting.

The authors are well-aware that the small sample size is the biggest problem of this work. That said, the article provides a model for their own work and we hope will encourage other researchers to expand upon these findings. This work can also encourage clinicians working with the chronically mentally ill to seek unique activities for their clients.

The Conclusions section is a very nice summary.

Author Response

(The authors gave the same response as above.)

Reviewer 3 Report

Comments and Suggestions for Authors

The manuscript "Effects of a One-Day Experiential Sheep-Rearing Experience on Motivation, Anxiety, and Frontal Lobe Brain Activity in Patients with Chronic Psychiatric Disorders: A Crossover Pilot Study" investigates the potential benefits of animal-assisted activities on individuals with chronic psychiatric disorders. It emphasizes the impact of a sheep-rearing experience on motivation and anxiety levels, along with changes in oxytocin and cortisol levels and brain activity in the prefrontal cortex.

Overall, the manuscript is well put together, and the study is relevant in assessing interventions to aid psychosocial rehabilitation. However, the writing could be more focused on delivering the message. The abstract could benefit from more specificity in the quantitative findings to strengthen its appeal to readers. While the manuscript contains some relevant studies on animal-assisted activities, it needs a critical analysis of the previous conclusions and how they relate to the current study's hypothesis. The paper would benefit from a rework of the introduction. A more thorough synthesis of existing research could enhance this section.

The authors have done a good job defining a thorough study protocol, providing the relevant patient and clinical characteristics, and ensuring that statistical analysis is pertinent. I commend the authors for a smart study design, though it was a limited cross-over pilot – I believe the conclusions drawn are relevant. I have an issue with the fact that the results and conclusions are not aptly tied to relevant literature or are not pertinently expanded. For example, the authors claim that the activation of DLPFC has not been previously observed in animal intervention studies and report this as a novel finding in psychiatric occupational therapy. I would expand on or highlight this finding (please include it in the abstract as well as a novel finding) and suggest possible neurobiological mechanisms associated with this activation, as this seems to be a clinically relevant finding. Some tables (Tables 1 and 5) can be moved to a supplementary section to keep the flow intact. A more detailed exploration of the study's implications and directions for future research would be beneficial.

While the study per se is conducted well and the results are interesting, the manuscript needs a deeper critical literature review both in the introduction and tying in the results and conclusions. In its current state, the manuscript is more descriptive, and I encourage the authors to expand the discussion on the results and compare them with existing studies to offer richer insights into the significance of the findings.

In summary, the manuscript presents an intriguing exploration of the rehabilitative potential of animal-assisted intervention for individuals with chronic psychiatric disorders. By addressing the suggested areas for improvement, the study could significantly contribute to psychiatric rehabilitation and animal-assisted therapy research. The additional enhancements, as described, would substantially strengthen the manuscript, and I recommend the study for publication with these changes.

Comments on the Quality of English Language

The quality of the English language is good, and I only recommend minor edits.

Author Response

(The authors gave the same response as above.)

Reviewer 4 Report

Comments and Suggestions for Authors

This manuscript entitled “Effects of a One-Day Experiential Sheep-Rearing Experience on Motivation, Anxiety, and Frontal Lobe Brain Activity in Patients with Chronic Psychiatric Disorders: A Crossover Pilot Study” described a pilot study investigating the effects of a one-day sheep-rearing experience on motivation, anxiety levels, and frontal lobe brain activity in individuals with chronic psychiatric disorders.  The study explores the intervention of sheep-rearing as a potential therapeutic activity for individuals with chronic psychiatric disorders. The authors evaluate multiple outcomes, including motivation levels measured using the Apathy Scale, changes in oxytocin and cortisol levels, and frontal lobe brain activity assessed using near-infrared spectroscopy, providing a holistic understanding of the intervention's effects.

Overall, the study presents an intriguing pilot study investigating the effects of a sheep-rearing experience on individuals with chronic psychiatric disorders, which would benefit the whole community.

Author Response

(The authors gave the same response as above.)
